**Data Availability Statement:** Raw RNA-Seq data have been deposited in the NCBI GEO sequence

# *Borrelia burgdorferi*, the Lyme disease spirochete, possesses genetically-encoded responses to doxycycline, but not to amoxicillin

Timothy C. Saylor[1‡], Timothy Casselli [2‡], Kathryn G. Lethbridge[1¤], Jessamyn P. Moore[1], Katie M. Owens[1], Catherine A. Brissette[2], Wolfram R. Zückert[3], Brian Stevenson [1,4]*

1 Department of Microbiology, Immunology and Molecular Genetics, University of Kentucky, School of Medicine, Lexington, Kentucky, United States of America, 2 Department of Biomedical Sciences, School of Medicine and Health Sciences, University of North Dakota, Grand Forks, North Dakota, United States of America, 3 Department of Microbiology, Molecular Genetics and Immunology, University of Kansas Medical Center, Kansas City, Kentucky, United States of America, 4 Department of Entomology, University of Kentucky, Lexington, Kentucky, United States of America

¤ Current address: Thermo Fisher, Grand Island, New York, United States of America
‡ TCS and TC contributed equally to this work and are to be considered co-equal first authors.
* brian.stevenson@uky.edu

## Abstract

Some species of bacteria respond to antibiotic stresses by altering their transcription profiles, in order to produce proteins that provide protection against the antibiotic. Understanding these compensatory mechanisms allows for informed treatment strategies, and could lead to the development of improved therapeutics. To this end, studies were performed to determine whether *Borrelia burgdorferi*, the spirochetal agent of Lyme disease, also exhibits genetically-encoded responses to the commonly prescribed antibiotics doxycycline and amoxicillin. After culturing for 24 h in a sublethal concentration of doxycycline, there were significant increases in a substantial number of transcripts for proteins that are involved with translation. In contrast, incubation with a sublethal concentration of amoxicillin did not lead to significant changes in levels of any bacterial transcript. We conclude that *B. burgdorferi* has a mechanism(s) that detects translational inhibition by doxycycline, and increases production of mRNAs for proteins involved with translation machinery in an attempt to compensate for that stress.

## Introduction

Lyme disease (Lyme borreliosis) is caused by infection by the spirochete *Borrelia burgdorferi* sensu lato (hereafter referred to as *B. burgdorferi*, for simplicity). Early manifestations include an expanding annular rash (erythema migrans) along with fever, body aches, and other "flu-like" symptoms. If untreated, more significant symptoms may be seen, including arthritis, meningitis, atrioventricular nodal block, or cardiac arrest [1–3]. This spirochete is sensitive to many types of antibiotics, and human Lyme disease is frequently treated with either

read archive database, and given accession number GSE197338.

**Funding:** BS, WRZ: R03 AI133056, National Institutes of Health. The funders had no role in study design, data collection and analysis, decision to publish, or preparation of the manuscript. CAB: U54GM128729 and 2P20GM104360-06A1, National Institutes of Health. The funders had no role in study design, data collection and analysis, decision to publish, or preparation of the manuscript.

**Competing interests:** The authors have declared that no competing interests exist.

doxycycline or amoxicillin [1, 2, 4–6]. Doxycycline inhibits bacterial translation, and amoxicillin inhibits assembly of cell wall peptidoglycan.

Some species of bacteria respond to the presence of antibiotics by modulating their gene and protein expression levels in efforts to overcome those stresses [7–12]. For examples, increasing production of efflux pumps or altering the relative expression levels of proteins involved with cell wall synthesis. Those observations raise the possibility that the Lyme disease spirochete may possess mechanisms that modify bacterial physiology in response to antibiotic therapies. Assessment of that possibility could inform prescribed antibiotics and dosages. Understanding these compensatory mechanisms allows for informed treatment strategies, and could lead to the development of new and/or improved therapeutics.

Exposing *B. burgdorferi* to sub-lethal levels of β-lactams may result in the spirochetes producing membrane protrusions or acquiring a spherical shape [13–18]. In other bacterial species, treatment with low levels of β-lactam antibiotics leads to weakening of the cell wall and cytoplasmic distortion due to osmotic influx of water [19–22]. However, there is a pervading hypothesis in the literature and among some physicians that β-lactam-induced "round bodies" are a genetically-encoded response by *B. burgdorferi* to avoid antibiotic killing [16–18, 23–34].

To address these points, we cultured *B. burgdorferi* in concentrations of doxycycline or amoxicillin that impaired, but did not completely prevent, bacterial replication. Bacteria were thus metabolically active, so changes could be interpreted as indicative of ongoing responses. To assess whether any physiological changes were due to genetically encoded processes, relative levels of mRNAs were compared for each condition.

## Material and methods

### Effects of antibiotic concentrations on replication of cultured *B. burgdorferi*

Strain B31-MI16, an infectious clone of *B. burgdorferi* type strain B31, was grown at 35°C to mid-exponential phase ($3 \times 10^7$ bacteria/ml) in liquid BSK-II medium [35, 36]. Triplicate aliquots of the culture were diluted 1:100 into fresh BSK-II that contained either no antibiotic, or 0.1, 0.2, or 0.4 μg/ml doxycycline or amoxicillin (Sigma). Bacterial numbers in each culture were then counted using a Petroff-Hauser counting chamber and dark field microscopy, marking time point 0. All cultures were counted every 24 hours for the first four days and on the seventh day. Antibiotic susceptibility assays were performed twice.

### Photomicrography

Aliquots of bacterial cultures were spread on glass slides, covered with coverslips, then visualized using dark field microscopy with a 40x objective lens. Images were recorded with a C-mounted Accu-scope Excelis HD camera using Captavision+ software. Bacterial lengths were determined by comparing their sizes against a reference stage micrometer, using Captavision + software. To quantify *B. burgdorferi* with membrane distortions after incubation for 24 h in 0.2 μg/ml of amoxicillin, bacteria in randomly selected fields were photographed, then assessed manually for presence of membrane perturbations. Due to variations in numbers of bacteria per field, 109 control bacteria and 110 amoxicillin-treated bacteria were assessed.

### Preparation of cultures for RNA sequencing

A mid-exponential phase ($3 \times 10^7$ bacteria/ml) 35°C culture of *B. burgdorferi* clone B31-MI16 was used as 1:100 inoculum into 18 separate tubes of 20ml BSK-II broth. Six cultures were not given any antibiotic, 6 received doxycycline to a final concentration of 0.2 μg/ml, and 6

cultures received amoxicillin to a final concentration of 0.2 μg/ml. After 3 hours incubation at 35˚C, 3 cultures of each condition were harvested by centrifugation for 15 min at 8200xG at 4˚C, then frozen at -80˚C. The remaining cultures were similarly harvested and frozen after 24 hours incubation at 35˚C. Frozen *B. burgdorferi* were shipped on dry ice to ACGT Inc. (https://www.acgtinc.com) for RNA processing and sequencing.

### RNA extraction and RNA sequencing (RNA-Seq)

Purification of RNA, preparation of libraries, and sequencing were performed by ACGT Inc. according to their standard protocols (https://www.acgtinc.com). Briefly, RNA was extracted from the bacterial pellets by using the Quick RNA-Microprep Kit (Zymo Research). RNA was evaluated with DeNovix and Nanodrop. An individual library was produced for each culture, using Zymo-Seq Ribofree Total RNA Library Kits (Zymo Research). Libraries were evaluated by Qubit and 2100 bioanalyzer to assess quality and quantity before sequencing. Sequencing was performed on Illumina NextSeq500 PE150. Runs were demultiplexed using bcl2fastq to obtain raw fastq files. Experimentally triplicated RNA-Seq produces robust data that do not require accompanying quantitative-reverse transcription PCR analyses [37].

### Bioinformatics

Analysis of transcriptome sequencing (RNA-Seq) data were performed in house, essentially described previously [38–40]. Briefly, adapters were removed from the sequencing reads by Trimmomatic [41]. The reads were aligned and counted with a transcriptome reference compiled from the *B. burgdorferi* strain B31-MI genome (RefSeq numbers AE000783 to AE000794 and AE001575 to AE001584) by using Salmon v1.5.2 [42]. Reads were normalized and differential expression analysis was conducted using DEseq2 [43]. Genes were considered to have significantly different expression at Fold-Change $\geq 2$, padj $\leq 0.05$, basemean $> 20$.

Data generated from RNA sequencing analyses were visualized with R v.4.0.1 (https://www.R-project.org/) using ggplot2 (https://doi.org/10.1007/978-3-319-24277-4) for MA plots, pie charts, and bar graphs.

Raw RNA-Seq data have been deposited in the NCBI GEO sequence read archive database, and given accession number GSE197338.

## Results and discussion

### Study design overview

To determine appropriate sublethal concentrations of antibiotics, an infectious clone of *B. burgdorferi* type strain B31 was cultured in liquid BSK-II medium that included various concentrations of either doxycycline or amoxicillin. Numbers of bacteria were counted daily over a course of 7 days, with inclusion of all motile and immobile spirochetes. Counting the number of organisms enabled determination of the effects of antibiotic treatment on completion of cell division. Under these culture conditions, this strain was completely inhibited from replicating by 0.4 μg/ml amoxicillin, while the minimum inhibitory concentration of doxycycline was greater than 0.4 μg/ml (Fig 1). Consistent with our findings, prior studies determined that minimum inhibitory and minimum bactericidal concentrations of doxycycline were 0.25–4 and 4–16 μg/ml, respectively, for Lyme disease borreliae [44]. Reported minimum inhibitory and minimum bactericidal concentrations of amoxicillin were 0.015–0.25 and 0.25–0.5 μg/ml, respectively [44]. In our investigations, concentrations of 0.2 μg/ml doxycycline and amoxicillin were found to substantially inhibit, but not eliminate, *B. burgdorferi* duplication (Fig 1).

## A. Doxycycline

## B. Amoxicillin

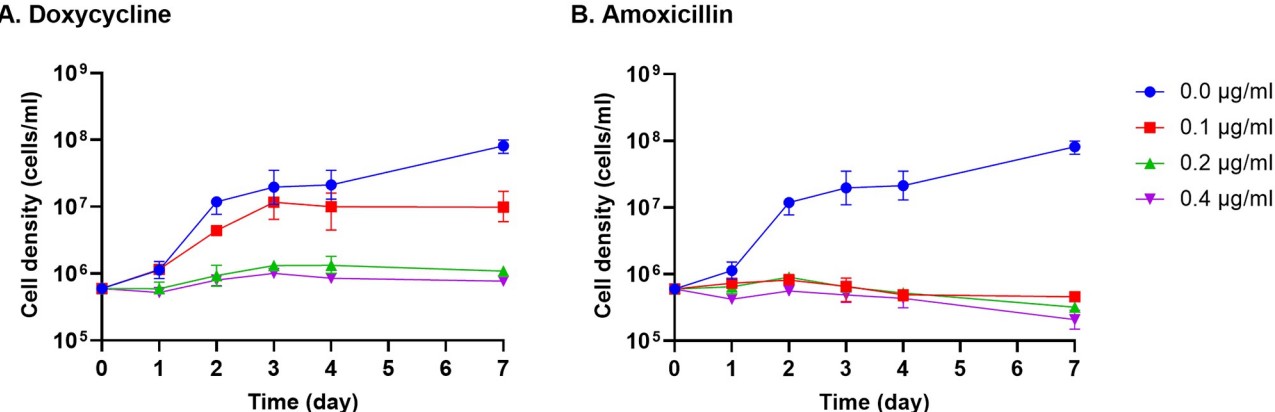

**Fig 1. Effects of antibiotics on *B. burgdorferi* replication rates.** (A) Doxycycline was added to freshly inoculated cultures at concentrations of 0.1 μg/mL, 0.2 μg/mL, and 0.4 μg/mL. (B) Amoxicillin was added to freshly inoculated cultures at concentrations 0.1 μg/mL, 0.2 μg/mL, and 0.4 μg/ml. Bacterial numbers were determined by microscopical examination with a Petroff-Hauser counting chamber after 1, 2, 3, 4 and 7 days of culture.

Those concentrations were designated "sublethal", and were subsequently tested for their effects on cell morphology and gene expression in *B. burgdorferi*.

Cultures were then grown to mid-exponential phase (approximately $3 \times 10^7$ bacteria / ml), diluted 1:100 into aliquots of fresh media, then either no antibiotic, or 0.2 μg/ml of either doxycycline or amoxicillin were added. Cultures were incubated at 35°C for either 3 or 24 hours prior to phenotype analysis. Longer time points were not examined, due to the increased possibility that substantial numbers of bacteria would die and their decaying RNA obscure results. To assess the effects of antibiotics on total gene expression, we took an unbiased approach using RNA sequencing (RNA-Seq). Effects of the antibiotics on bacterial morphologies were assessed by darkfield microscopy.

Under the sublethal concentrations of antibiotics used in our studies, bacteria continued to move, elongate, and divide, indicating that the spirochetes were metabolically active (Fig 1 and discussion below). These conditions allowed us to differentiate biological responses to antibiotics from experimental artefacts from dead and/or dying bacteria. On the other hand, two previous transcriptomic analyses of *B. burgdorferi* cultivated in antibiotics used concentrations of 50 μg/ml doxycycline [45, 46] or 50 μg/ml amoxicillin [45] for 5 days before RNA analyses. Those levels are 12 to 100-times greater than the minimum bactericidal concentrations [44]. Neither of those studies examined the physiology of *B. burgdorferi* during incubation under those conditions [45, 46].

## Doxycycline induced gene expression changes associated with protein translation

Exposure of *B. burgdorferi* to 0.2 μg/ml doxycycline led to an initial significant decrease in expression of 36 genes after three hours compared to control cells without antibiotics (Fold-Change $\geq 2$, padj $\leq 0.05$, basemean $> 20$), while no genes were significantly increased at this timepoint (Fig 2A; Table 1; S1 Table). Differentially expressed genes (DEGs) included those involved in protein translation, DNA replication/repair, cell motility, and carbohydrate metabolism, however only a small number of genes ($\leq 7$) from each pathway were affected (Fig 2B and 2C). Due to the low number of differentially expressed genes and the diversity of predicted functions, it is possible that these differences reflect nonspecific mRNA turnover differences in the presence of doxycycline. There are no obvious benefits to reducing levels of those transcripts.

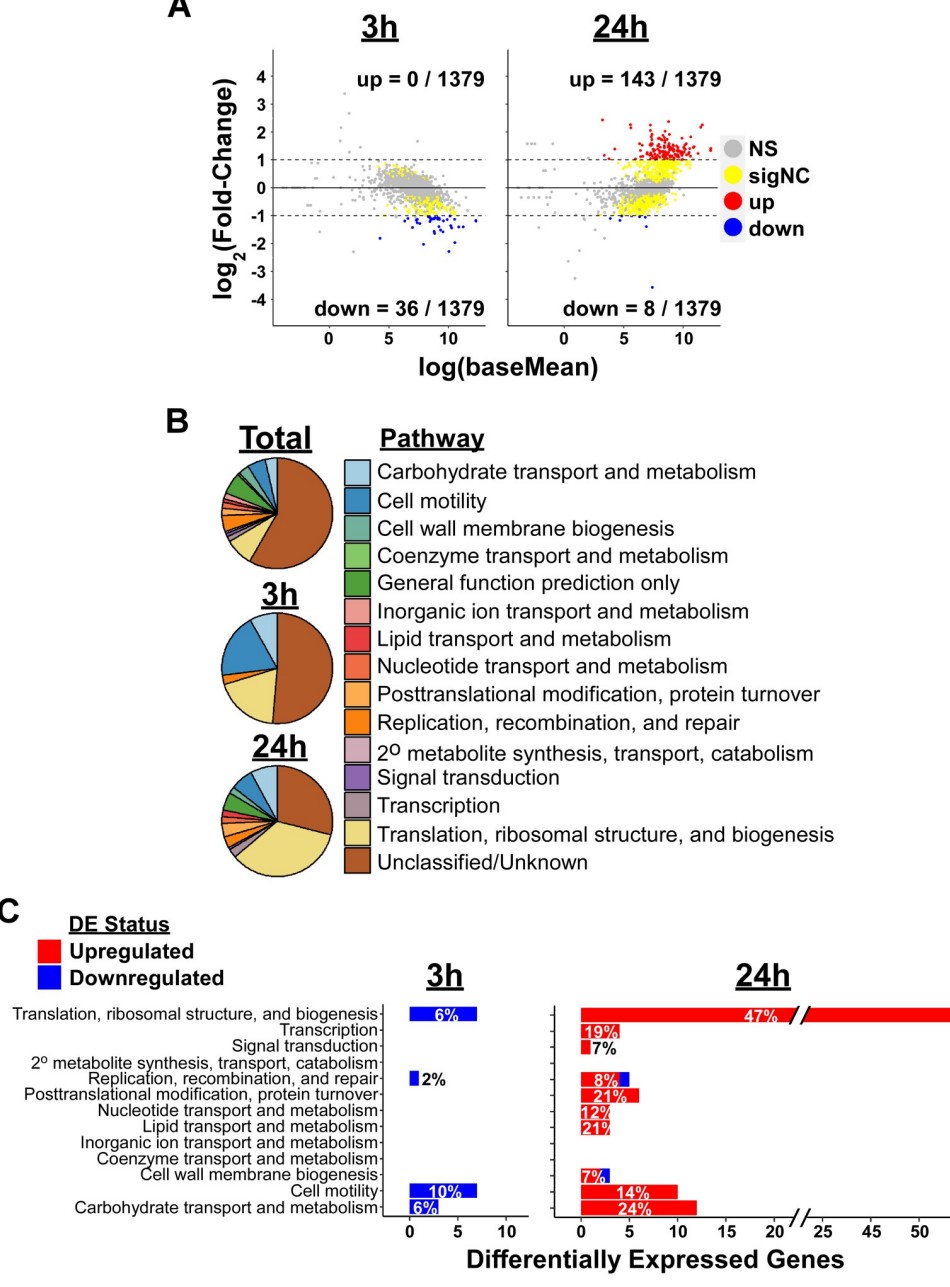

**Fig 2. Doxycycline induced gene expression changes associated with protein translation.** (A) Fold change versus expression strength for all detectable genes after 3 or 24 hours doxycycline treatment compared to untreated controls. Red (increased) and blue (decreased) dots represent genes with significantly different levels in treated vs. control bacteria ($\alpha = 0.05$, $\log_2$(fold-change) > 1). Yellow dots represent significantly different expression ($\alpha = 0.05$) without meeting our fold-change cutoff for differential expression ("sigNC"). Gray dots represent genes that were not significantly different between treatment and control bacteria ("NS"). Numbers of significantly upregulated (up) and downregulated (down) genes are shown as proportions of all detectable genes. (B) Clusters of Orthologous Genes (COG) pathways displayed as proportion of all detectable genes ("Total") compared to differentially expressed genes after 3h or 24h of doxycycline treatment [47]. (C) Stacked bar graph showing the number of increased (red) and decreased (blue) genes in each COG pathway at 3h and 24h timepoints. Percentage of genes in each pathway that were differentially expressed is stated within each bar. Note: Unclassified and general function prediction not shown.

**Table 1. Differentially expressed genes in doxycycline treated *B. burgdorferi* versus untreated controls.**

| locus | Description | COG pathway | log₂(fold change)[a] 3h | log₂(fold change)[a] 24h |
|---|---|---|---|---|
| BB_0691 | elongation factor G (fusA) | Translation ribosomal structure and biogenesis | -1.01 | 1.52 |
| BB_0786 | 50S ribosomal protein L25/general stress protein Ctc | Translation ribosomal structure and biogenesis | -1.04 | 1.30 |
| BB_0479 | 50S ribosomal protein L4 (rplD) | Translation ribosomal structure and biogenesis | -1.08 | 1.40 |
| BB_0690 | neutrophil activating protein A (napA) | Replication recombination and repair | -1.08 | 2.02 |
| BB_0055 | triosephosphate isomerase (tpiA) | Carbohydrate transport and metabolism | -1.08 | 1.38 |
| BB_0328 | family 5 extracellular solute-binding protein | Unclassified | -1.10 | NS |
| BB_0428 | hypothetical protein | Unclassified | -1.10 | 1.12 |
| BB_0330 | peptide ABC transporter substrate-binding protein | Unclassified | -1.11 | NS |
| BB_J09 | outer surface protein D (ospD) | Unclassified | -1.12 | NS |
| BB_0383 | basic membrane protein A (bmpA) | Cell motility | -1.13 | NS |
| BB_0603 | integral outer membrane protein p66 (p66) | Unclassified | -1.13 | NS |
| BB_0715 | cell division protein FtsA (ftsA) | Unclassified | -1.14 | NS |
| BB_0651 | protein translocase subunit YajC | Cell motility | -1.14 | 1.57 |
| BB_0034 | outer membrane protein P13 | Unclassified | -1.14 | 1.77 |
| BB_0387 | 30S ribosomal protein S12 (rpsL) | Translation ribosomal structure and biogenesis | -1.15 | 1.11 |
| BB_A15 | outer surface protein A (ospA) | Unclassified | -1.18 | 1.33 |
| BB_0337 | enolase (eno) | Carbohydrate transport and metabolism | -1.19 | NS |
| BB_0650 | hypothetical protein | Unclassified | -1.20 | 1.58 |
| BB_A16 | outer surface protein B (ospB) | Unclassified | -1.20 | 1.41 |
| BB_0293 | flagellar basal body rod protein FlgC (flgC) | Cell motility | -1.21 | 1.17 |
| BB_0396 | 50S ribosomal protein L33 (rpmG) | Translation ribosomal structure and biogenesis | -1.26 | 1.91 |
| BB_0090 | V-type ATP synthase subunit K | Unclassified | -1.29 | 1.06 |
| BB_0385 | basic membrane protein D (bmpD) | Cell motility | -1.36 | NS |
| BB_A74 | outer membrane porin OMS28 (osm28) | Cell motility | -1.40 | NS |
| BB_0147 | flagellin (flaB) | Cell motility | -1.41 | 1.22 |
| BB_r05 | rna13 gene (16S) | Unclassified | -1.41 | NS |
| BB_0054 | protein-export membrane protein SecG (secG) | Cell motility | -1.43 | 1.26 |
| BB_r02 | rna8 gene (23S rrlA) | Unclassified | -1.44 | 1.18 |
| BB_0243 | glycerol-3-phosphate dehydrogenase | Unclassified | -1.52 | NS |
| BB_0240 | glycerol uptake facilitator | Carbohydrate transport and metabolism | -1.57 | NS |
| BB_0386 | 30S ribosomal protein S7 (rpsG) | Translation ribosomal structure and biogenesis | -1.58 | NS |
| BB_0465 | hypothetical protein | Unclassified | -1.64 | 1.91 |
| BB_r01 | rna7 gene = BB r01 | Unclassified | -1.81 | 1.38 |
| BB_r04 | rna10 gene (23S rrlB) | Unclassified | -1.97 | 1.40 |
| BB_0631 | hypothetical protein | Unclassified | -2.03 | 1.76 |
| BB_0241 | glycerol kinase (glpK) | Unclassified | -2.29 | NS |
| rnaseP | rnaseP | Unclassified | NS | 2.42 |
| BB_0188 | 50S ribosomal protein L20 (rplT) | Translation ribosomal structure and biogenesis | NS | 2.37 |
| BB_P40 | hypothetical protein | Unclassified | NS | 2.26 |
| BB_0649 | chaperonin GroEL (groL) | Posttranslational modification protein turnover chaperones | NS | 2.25 |
| bsrW | bsrW | Unclassified | NS | 2.25 |
| BB_B29 | PTS system transporter subunit IIBC | Carbohydrate transport and metabolism | NS | 2.17 |
| BB_0614 | hypothetical protein | Unclassified | NS | 2.14 |
| BB_0741 | chaperonin GroS (groS) | Posttranslational modification protein turnover chaperones | NS | 2.05 |
| BB_0501 | 30S ribosomal protein S11 (rpsK) | Translation ribosomal structure and biogenesis | NS | 1.88 |
| BB_0780 | 50S ribosomal protein L27 (rpmA) | Translation ribosomal structure and biogenesis | NS | 1.82 |

(*Continued*)

**Table 1.** (*Continued*)

| locus | Description | COG pathway | log$_2$(fold change)[a] | |
| --- | --- | --- | --- | --- |
| | | | **3h** | **24h** |
| BB_0445 | fructose-bisphosphate aldolase (fbaA) | Carbohydrate transport and metabolism | NS | 1.82 |
| BB_0393 | 50S ribosomal protein L11 (rplK) | Translation ribosomal structure and biogenesis | NS | 1.79 |
| BB_A62 | 6.6 kDa lipoprotein (lp6.6) | Unclassified | NS | 1.78 |
| BB_0503 | 50S ribosomal protein L17 (rplQ) | Translation ribosomal structure and biogenesis | NS | 1.73 |
| BB_0405 | hypothetical protein | Unclassified | NS | 1.73 |
| BB_0778 | 50S ribosomal protein L21 (rplU) | Translation ribosomal structure and biogenesis | NS | 1.72 |
| BB_0482 | 30S ribosomal protein S19 (rpsS) | Translation ribosomal structure and biogenesis | NS | 1.71 |
| BB_0776 | hypothetical protein | Unclassified | NS | 1.69 |
| BB_0559 | PTS system glucose-specific transporter subunit IIA | Carbohydrate transport and metabolism | NS | 1.61 |
| BB_O27 | protein BdrN (bdrN) | Unclassified | NS | 1.58 |
| BB_0238 | hypothetical protein | General function prediction only | NS | 1.58 |
| BB_0057 | glyceraldehyde 3-phosphate dehydrogenase (gap) | Carbohydrate transport and metabolism | NS | 1.57 |
| BB_0489 | 50S ribosomal protein L24 (rplX) | Translation ribosomal structure and biogenesis | NS | 1.55 |
| BB_0113 | 30S ribosomal protein S18 (rpsR) | Translation ribosomal structure and biogenesis | NS | 1.55 |
| BB_0781 | GTPase Obg | General function prediction only | NS | 1.53 |
| BB_0779 | hypothetical protein | Translation ribosomal structure and biogenesis | NS | 1.52 |
| BB_0189 | 50S ribosomal protein L35 (rpmI) | Translation ribosomal structure and biogenesis | NS | 1.52 |
| BB_0488 | 50S ribosomal protein L14 (rplN) | Translation ribosomal structure and biogenesis | NS | 1.52 |
| BB_0802 | ribosome-binding factor A (rbfA) | Translation ribosomal structure and biogenesis | NS | 1.51 |
| BB_0392 | 50S ribosomal protein L1 (rplA) | Translation ribosomal structure and biogenesis | NS | 1.50 |
| BB_0502 | DNA-directed RNA polymerase subunit alpha (rpoA) | Transcription | NS | 1.50 |
| BB_0114 | single-stranded DNA-binding protein | Replication recombination and repair | NS | 1.50 |
| BB_0366 | aminopeptidase | Unclassified | NS | 1.49 |
| BB_0115 | 30S ribosomal protein S6 | Translation ribosomal structure and biogenesis | NS | 1.49 |
| BB_0805 | polyribonucleotide nucleotidyltransferase | Translation ribosomal structure and biogenesis | NS | 1.47 |
| BB_0485 | 50S ribosomal protein L16 (rplP) | Translation ribosomal structure and biogenesis | NS | 1.46 |
| BB_0695 | 30S ribosomal protein S16 (rpsP) | Translation ribosomal structure and biogenesis | NS | 1.46 |
| BB_0500 | 30S ribosomal protein S13 (rpsM) | Translation ribosomal structure and biogenesis | NS | 1.45 |
| BB_0390 | 50S ribosomal protein L7/L12 (rplL) | Translation ribosomal structure and biogenesis | NS | 1.45 |
| BB_0504 | ribonuclease Y | General function prediction only | NS | 1.44 |
| BB_0476 | elongation factor Tu (tuf) | Translation ribosomal structure and biogenesis | NS | 1.44 |
| BB_0348 | pyruvate kinase (pyk) | Carbohydrate transport and metabolism | NS | 1.43 |
| BB_0699 | 50S ribosomal protein L19 (rplS) | Translation ribosomal structure and biogenesis | NS | 1.43 |
| BB_0558 | phosphoenolpyruvate-protein phosphatase (ptsP) | Carbohydrate transport and metabolism | NS | 1.40 |
| BB_0128 | cytidylate kinase (cmk) | Nucleotide transport and metabolism | NS | 1.38 |
| BB_0785 | septation protein SpoVG (spoVG) | Cell wall membrane biogenesis | NS | 1.38 |
| BB_0478 | 50S ribosomal protein L3 (rplC) | Translation ribosomal structure and biogenesis | NS | 1.38 |
| BB_0493 | 50S ribosomal protein L6 | Translation ribosomal structure and biogenesis | NS | 1.37 |
| BB_0069 | aminopeptidase II | Unclassified | NS | 1.37 |
| BB_0394 | transcription termination/antitermination factor (nusG) | Transcription | NS | 1.35 |
| BB_0283 | flagellar hook protein FlgE (flgE) | Cell motility | NS | 1.34 |
| BB_0494 | 50S ribosomal protein L18 (rplR) | Translation ribosomal structure and biogenesis | NS | 1.34 |
| BB_0495 | 30S ribosomal protein S5 (rpsE) | Translation ribosomal structure and biogenesis | NS | 1.34 |
| BB_0229 | 50S ribosomal protein L31 type B (rpmE) | Translation ribosomal structure and biogenesis | NS | 1.33 |
| BB_0269 | ATP-binding protein | Unclassified | NS | 1.33 |
| BB_0112 | 50S ribosomal protein L9 (rplI) | Translation ribosomal structure and biogenesis | NS | 1.32 |

(*Continued*)

**Table 1.** (Continued)

| locus | Description | COG pathway | log$_2$(fold change)[a] | |
|---|---|---|---|---|
| | | | **3h** | **24h** |
| BB_0683 | 3-hydroxy-3-methylglutaryl-CoA synthase | Lipid transport and metabolism | NS | 1.32 |
| BB_0087 | L-lactate dehydrogenase | Unclassified | NS | 1.30 |
| BB_0477 | 30S ribosomal protein S10 (rpsJ) | Translation ribosomal structure and biogenesis | NS | 1.30 |
| BB_0047 | hypothetical protein | Unclassified | NS | 1.30 |
| BB_0355 | transcription factor | Transcription | NS | 1.30 |
| BB_0277 | flagellar motor switch protein FliN (fliN) | Cell motility | NS | 1.29 |
| BB_0481 | 50S ribosomal protein L2 (rplB) | Translation ribosomal structure and biogenesis | NS | 1.29 |
| BB_0658 | 23-bisphosphoglycerate-dependent phosphoglycerate mutase | Carbohydrate transport and metabolism | NS | 1.29 |
| BB_0570 | chemotaxis response regulator | Signal transduction mechanisms | NS | 1.28 |
| BB_0436 | DNA gyrase subunit B (gyrB) | Replication recombination and repair | NS | 1.28 |
| BB_0483 | 50S ribosomal protein L22 (rplV) | Translation ribosomal structure and biogenesis | NS | 1.27 |
| BB_0056 | phosphoglycerate kinase (pgk) | Carbohydrate transport and metabolism | NS | 1.26 |
| BB_0841 | arginine deiminase (arcA) | Unclassified | NS | 1.26 |
| BB_0539 | hypothetical protein | General function prediction only | NS | 1.26 |
| BB_0694 | signal recognition particle protein (ffh) | Cell motility | NS | 1.26 |
| BB_0557 | phosphocarrier protein HPr | Carbohydrate transport and metabolism | NS | 1.24 |
| BB_0777 | adenine phosphoribosyltransferase (apt) | Nucleotide transport and metabolism | NS | 1.24 |
| BB_0127 | 30S ribosomal protein S1 | Translation ribosomal structure and biogenesis | NS | 1.24 |
| BB_0122 | elongation factor Ts (tsf) | Translation ribosomal structure and biogenesis | NS | 1.22 |
| BB_0789 | ATP-dependent zinc metalloprotease FtsH | Posttranslational modification protein turnover chaperones | NS | 1.21 |
| BB_0492 | 30S ribosomal protein S8 (rpsH) | Translation ribosomal structure and biogenesis | NS | 1.21 |
| BB_0704 | acyl carrier protein (acpP) | Lipid transport and metabolism | NS | 1.21 |
| BB_0104 | periplasmic serine protease DO | Posttranslational modification protein turnover chaperones | NS | 1.21 |
| BB_0426 | nucleoside 2-deoxyribosyltransferase superfamily protein | Function unknown | NS | 1.20 |
| BB_0027 | hypothetical protein | Unclassified | NS | 1.20 |
| BB_0123 | 30S ribosomal protein S2 (rpsB) | Translation ribosomal structure and biogenesis | NS | 1.20 |
| BB_B19 | outer surface protein C (ospC) | Unclassified | NS | 1.18 |
| BB_0727 | phosphofructokinase | Carbohydrate transport and metabolism | NS | 1.18 |
| BB_0697 | ribosome maturation factor RimM (rimM) | Translation ribosomal structure and biogenesis | NS | 1.18 |
| BB_B22 | guanine/xanthine permease | General function prediction only | NS | 1.17 |
| BB_0061 | thioredoxin (trx) | Posttranslational modification protein turnover chaperones | NS | 1.17 |
| BB_0338 | 30S ribosomal protein S9 (rpsI) | Translation ribosomal structure and biogenesis | NS | 1.17 |
| BB_0499 | 50S ribosomal protein L36 (rpmJ) | Translation ribosomal structure and biogenesis | NS | 1.16 |
| BB_0121 | ribosome recycling factor (frr) | Translation ribosomal structure and biogenesis | NS | 1.16 |
| BB_0744 | p83/100 antigen (p83/100) | Unclassified | NS | 1.16 |
| BB_0172 | von Willebrand factor type A domain-containing protein | Function unknown | NS | 1.16 |
| BB_0615 | 30S ribosomal protein S4 (rpsD) | Translation ribosomal structure and biogenesis | NS | 1.15 |
| BB_0391 | 50S ribosomal protein L10 | Translation ribosomal structure and biogenesis | NS | 1.14 |
| BB_0190 | translation initiation factor IF-3 (infC) | Translation ribosomal structure and biogenesis | NS | 1.14 |
| BB_0059 | CBS domain-containing protein | General function prediction only | NS | 1.14 |
| BB_0698 | tRNA (guanine-N(1)-)-methyltransferase (trmD) | Translation ribosomal structure and biogenesis | NS | 1.13 |
| BB_0435 | DNA gyrase subunit A (gyrA) | Replication recombination and repair | NS | 1.13 |
| BB_0735 | rare lipoprotein A | Cell wall membrane biogenesis | NS | 1.12 |
| BB_0120 | isoprenyl transferase (uppS) | Lipid transport and metabolism | NS | 1.12 |
| BB_0518 | chaperone protein DnaK (dnaK) | Posttranslational modification protein turnover chaperones | NS | 1.12 |
| BB_0684 | isopentenyl-diphosphate delta-isomerase (fni) | Unclassified | NS | 1.12 |

*(Continued)*

**Table 1.** (Continued)

| locus | Description | COG pathway | log$_2$(fold change)$^a$ | |
|-------|-------------|-------------|:---:|:---:|
| | | | **3h** | **24h** |
| BB_0487 | 30S ribosomal protein S17 (rpsQ) | Translation ribosomal structure and biogenesis | NS | 1.11 |
| BB_0231 | hypothetical protein | Function unknown | NS | 1.10 |
| BB_0020 | diphosphate—fructose-6-phosphate 1-phosphotransferase | Carbohydrate transport and metabolism | NS | 1.10 |
| BB_0339 | 50S ribosomal protein L13 (rplM) | Translation ribosomal structure and biogenesis | NS | 1.09 |
| BB_0144 | glycine/betaine ABC transporter substrate-binding protein | Unclassified | NS | 1.09 |
| BB_0588 | MTA/SAH nucleosidase | Nucleotide transport and metabolism | NS | 1.08 |
| BB_0125 | hypothetical protein | Unclassified | NS | 1.07 |
| BB_0230 | transcription termination factor Rho (rho) | Transcription | NS | 1.06 |
| BB_0497 | 50S ribosomal protein L15 (rplO) | Translation ribosomal structure and biogenesis | NS | 1.06 |
| BB_0611 | ATP-dependent Clp protease proteolytic subunit (clpP) | Cell motility | NS | 1.06 |
| BB_0842 | ornithine carbamoyltransferase (argF) | Unclassified | NS | 1.05 |
| BB_0171 | hypothetical protein | General function prediction only | NS | 1.04 |
| BB_0533 | protein PhnP | General function prediction only | NS | 1.04 |
| BB_0491 | 30S ribosomal protein S14 (rpsN) | Translation ribosomal structure and biogenesis | NS | 1.03 |
| BB_0652 | protein translocase subunit SecD (secD) | Cell motility | NS | 1.01 |
| BB_0281 | motility protein A (motA) | Cell motility | NS | 1.01 |
| BB_0484 | 30S ribosomal protein S3 (rpsC) | Translation ribosomal structure and biogenesis | NS | 1.01 |
| BB_0070 | hypothetical protein | Function unknown | NS | 1.00 |
| BB_0067 | peptidase | Unclassified | NS | 1.00 |
| BB_0460 | lipoprotein | Unclassified | NS | -1.01 |
| BB_J46 | hypothetical protein | Replication recombination and repair | NS | -1.02 |
| BB_M37 | protein BppC (bppC) | Unclassified | NS | -1.04 |
| BB_0546 | hypothetical protein | Cell wall membrane biogenesis | NS | -1.06 |
| BB_M14 | hypothetical protein | Unclassified | NS | -1.10 |
| BB_F14 | hypothetical protein | Unclassified | NS | -1.17 |
| BB_L41 | hypothetical protein | Unclassified | NS | -1.40 |
| tmRNA | tmRNA | Unclassified | NS | -3.57 |

Fold changes are expressed as log$_2$. NS = not a significant change

$^a$"NS" denotes not significantly different ($\alpha$ = 0.05; log$_2$(fold change)>1)

After 24 hours of doxycycline treatment, microscopical examination showed that *B. burgdorferi* were motile and were, therefore, metabolically active. RNA-Seq analyses at that time point revealed that 151 genes were differentially expressed (143 upregulated, 8 downregulated) compared to control cells (Fig 2A, Table 1, and S1 Table). Notably, a plurality of differentially expressed genes (53/151 DEGs; 35%) are involved in protein synthesis, all of which were upregulated in the treatment group (Fig 2B and 2C). These genes account for nearly half (47%) of all genes annotated as belonging to the translation, ribosomal structure, and biogenesis pathway (Fig 2C) [47]. These gene expression changes indicate that *B. burgdorferi* possesses a genetically-encoded mechanism(s) that attempts to overcome ribosome impairment, which is focused on enhanced production of mRNAs for components of translation.

The most common mechanism of bacterial resistance to tetracyclines is through efflux pumps that export the antibiotic from the cell [48]. While *B. burgdorferi* naturally encodes an efflux pump, BesCAB [49], levels of *besCAB* mRNA were not affected by presence of doxycycline (Table 1 and S1 Table). *B. burgdorferi* does not encode homologues of any known enzyme

that could modify doxycycline [50, 51], so that possible mechanism is unlikely to affect survival in the presence of the antibiotic.

Notably, expression of *napA* increased after 24h exposure to doxycycline (Table 1). This transcript encodes a periplasmic protein (also called BicA) that can bind copper and manganese, and associates with cell wall peptidoglycan [52–54]. The predicted sequence of NapA is similar to the Dps proteins of other bacterial species, which are involved with protecting DNA from stresses [55], although borrelia NapA lacks the Dps sequences that are involved with DNA-binding [52]. NapA derives its name from **n**eutrophil **a**ttracting **p**rotein **A**, and has been demonstrated to enhance immune responses [52, 54, 56–58]. It remains to be seen whether differential expressionof NapA occurs during doxycycline treatment in the context of mammalian infection.

As noted above, two other research groups have published results of RNA-Seq analyses of *B. burgdorferi* that were incubated for 5 days in 50 μg/ml doxycycline [45, 46]. The doxycycline concentration used in those studies was many times greater than what we and others found to inhibit *B. burgdorferi* replication in culture [44]. Although Feng et al. [45] described *B. burgdorferi* that had been incubated in 50 μg/ml doxycycline as "persisters", those researchers did not assess the viability of the bacteria that were used for RNA-Seq analysis. We also point out that the accepted definition of bacterial persistence cannot be applied to bacteriostatic antibiotics such as doxycycline, since the nature of those antibiotics does not directly kill bacteria [59]. The high dosages used by Feng at al. and Caskey et al. may explain why there is very little overlap between their results, despite both using essentially the same culture conditions [45, 46]. Feng et al. reported significant (> 2-fold) increases of 35 transcripts and decreases of 33 transcripts, encoding a broad range of functions [45]. In contrast, Caskey et al. [46] noted increases of 20 transcripts, while 40 transcripts were found to be downregulated. Many of the downregulated transcripts were of various plasmid encoded outer surface proteins. Of the 20 upregulated transcripts reported by Caskey et al., all but one came from the Lyme spirochete's resident cp32 prophages [60]. This is unlike the broadranging transcript groups reported to be upregulated by Feng et al. It is not clear whether the Caskey et al. results can be interpreted to imply anything about the native prophage's responses to doxycycline stress, since the vast majority of prophage genes were not affected. The increased transcripts encode portal proteins of four different cp32 bacteriophages, and three different Erp lipoproteins that localize to the bacteria's outer surface, are not predicted to be components of the bacteriophage particle, and do not possess functions relevant to survival in doxycycline [60–64].

Caskey et al. found that some bacteria had survived incubation for 5 days in 50μg/ml doxycycline, and resumed growth when subcultured in fresh medium without antibiotic or injected into mice [46]. That result is consistent with our observations of continued bacterial motility when exposed to 0.2 μg/ml doxycycline. As with other bacterial species, tetracyclines are bacteriostatic to *B. burgdorferi*, rather than overtly bactericidal [65].

## Amoxicillin resulted in morphological changes, but not changes in gene expression

Amoxicillin is a β-lactam, which inhibits cell wall production. In contrast to doxycycline, exposure for 3 or 24 hours to 0.2 μg/ml amoxicillin did not result in significant changes to any transcript, even without a fold-change cutoff for differential expression designation (Fig 3 and S1 Table). The previous study by Feng et al. [45] reported that 5 days incubation in 50 μg/ml amoxicillin resulted in their detection of significant increases in 41 mRNAs of a range of functions, but none of which encode proteins involved with cell wall or membrane synthesis or

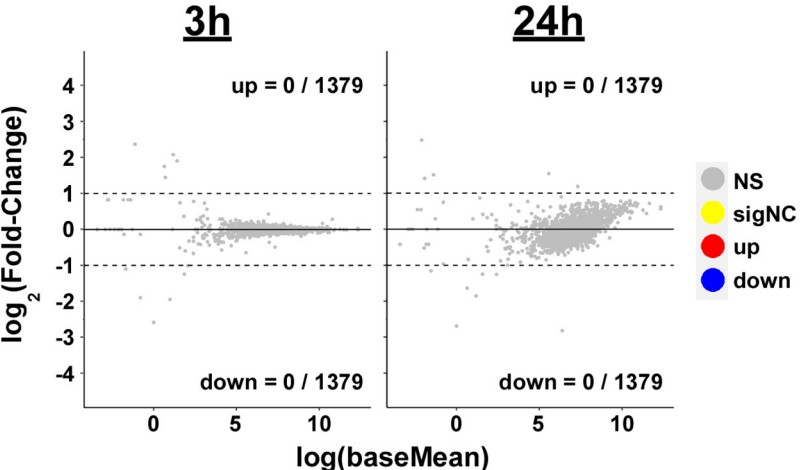

**Fig 3. Amoxicillin did not induce gene expression changes.** Fold change versus expression strength for all detectable genes after 3 or 24 hours amoxicillin treatment compared to untreated controls. No genes were significantly different between treatment and control groups (α = 0.05), as indicated by gray dots ("NS").

remodeling. As noted above, Feng et al. did not assess bacterial viability before their RNA-Seq analyses.

The absence of cell wall-directed responses to amoxicillin suggests that *B. burgdorferi* may lack a mechanism to assess cell wall integrity. While many bacterial species recycle peptidoglycan components as they grow in size, *B. burgdorferi* lacks such an ability, and instead sheds remnants of cell wall remodeling into the environment [66]. Together, these suggest that *B. burgdorferi* transports peptidoglycan components into the periplasm to build its cell wall as it grows in length, while "assuming" that the cell wall is being assembled correctly.

Examination under the microscope revealed that amoxicillin-treated *B. burgdorferi* displayed evidence of membrane swelling (Fig 4). After 24 h in the antibiotic, microscopical examination of randomly selected bacteria showed membrane distensions in 49/110 (44.6%)

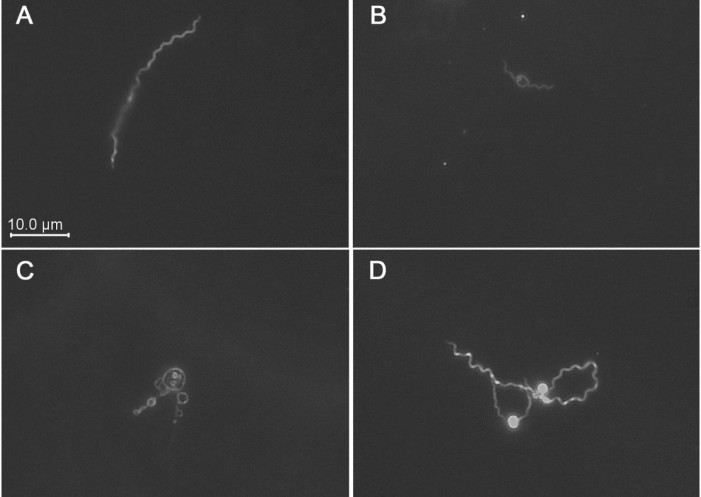

**Fig 4. Photomicrographs of representative *B. burgdorferi* from (A) control, or (B, C, and D) amoxicillin-treated cultures after 24h incubation.** All fields are shown at the same relative magnification. Imaged with a 40x objective lens and darkfield illumination.

of amoxicillin-treated spirochetes, as compared to 6/109 (5.5%) of control *B. burgdorferi*. Those bacteria were comparable in shape to the so-called "round bodies" or "cysts" that have previously been described upon treatment of cultured *B. burgdorferi* with sublethal concentrations of β-lactams [13, 14, 16, 18, 33]. However, our transcriptomic analyses indicate that the amoxicillin-induced morphological changes were not genetically encoded. Instead, the observed membrane swellings were probably results of water diffusing into the cytoplasm and expanding the inner membrane that was no longer constrained by an intact cell wall. Similar osmotically-induced spheroplasts can be generated in other bacterial species through β-lactam induced weakening of their cell walls [19–22]. Although β-lactam derived spheroplasts of *B. burgdorferi* are evidently not biologically relevant to these bacteria in nature, such experimentally-derived structures can be useful for investigations of membrane functions [21, 22].

## Conclusions

In nature, the Lyme disease spirochete exists only within vertebrates or ticks. In those environments, it is unlikely that *B. burgdorferi* would routinely encounter molds that produce β-lactam antibiotics and thus would not have been under pressure to evolve escape strategies. The evident inability of *B. burgdorferi* to respond to amoxicillin's inhibition of cell wall synthesis supports that hypothesis. Our data also suggest that *B. burgdorferi* does not naturally encounter other conditions that block peptidoglycan synthesis, and thus has not evolved mechanisms to respond to such a stress.

In contrast, *B. burgdorferi* evidently possesses a mechanism(s) that detects the impairment of translation due to doxycycline, and attempts to overcome that inhibition by increasing expression of genes involved with translation. Tetracyclines are synthesized in nature by actinomycete bacteria, which are predominantly soil microbes and are therefore unlikely to be encountered by *B. burgdorferi* in nature [67]. It remains to be seen whether other methods that inhibit translation yield similar effects. Nonetheless, the response of *B. burgdorferi* raises questions about where these spirochetes encounter translational impairment in their natural tick-vertebrate infectious cycle. One possibility is in the midgut of an unfed tick, where *B. burgdorferi* is starved for amino acids; accumulation of mRNAs for producing translation-associated proteins might allow rapid production of those proteins when the tick begins feeding on nutrient-rich blood. Further studies of Lyme disease spirochete physiology during its infectious cycle can help solve this question.

Taken together, our studies found that *B. burgdorferi* demonstrates distinct responses to different antibiotics. While it may be that *B. burgdorferi* within vertebrate tissues activate regulatory pathways that are not observed in culture, and thereby adapt to tolerate antibiotics, we also note that there is no direct evidence to support such hypothetical mechanisms. Importantly, neither our studies or those of Feng at al. or Caskey et al. [45, 46] directly addressed the efficacy of doxycycline or amoxycillin for treatment of Lyme disease in humans, as those treatments have been determined empirically. Rather, these insights shed light on the feedback mechanisms to environmental stresses by *B. burgdorferi*, and could lead to the development of novel therapeutic treatments for this important pathogen.

## Supporting information

**S1 Table. All results for doxycycline after 3h and 24h, and all results for amoxicillin after 3h and 24h.**
(XLSX)

## Acknowledgments

We thank Tatiana Castro-Padovani, Nerina Jusufovic, and Andrew Krusenstjerna for helpful comments on these studies and the manuscript.

## Author Contributions

**Conceptualization:** Wolfram R. Zückert, Brian Stevenson.

**Data curation:** Timothy C. Saylor, Timothy Casselli, Catherine A. Brissette, Wolfram R. Zückert, Brian Stevenson.

**Formal analysis:** Timothy C. Saylor, Timothy Casselli, Jessamyn P. Moore, Katie M. Owens, Catherine A. Brissette, Wolfram R. Zückert, Brian Stevenson.

**Funding acquisition:** Catherine A. Brissette, Wolfram R. Zückert, Brian Stevenson.

**Investigation:** Timothy C. Saylor, Kathryn G. Lethbridge, Jessamyn P. Moore, Katie M. Owens.

**Methodology:** Timothy C. Saylor, Timothy Casselli, Wolfram R. Zückert, Brian Stevenson.

**Project administration:** Catherine A. Brissette, Wolfram R. Zückert, Brian Stevenson.

**Resources:** Catherine A. Brissette, Wolfram R. Zückert, Brian Stevenson.

**Supervision:** Brian Stevenson.

**Validation:** Timothy Casselli, Jessamyn P. Moore, Katie M. Owens.

**Visualization:** Timothy Casselli, Brian Stevenson.

**Writing – original draft:** Brian Stevenson.

**Writing – review & editing:** Timothy C. Saylor, Timothy Casselli, Catherine A. Brissette, Wolfram R. Zückert, Brian Stevenson.

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
