## [Decision Letter · Decision Letter 0]

7 Jun 2022

PONE-D-22-10059Borrelia burgdorferi, the Lyme disease spirochete, possesses genetically-encoded responses to doxycycline, but not to amoxicillinPLOS ONE

Dear Dr. Stevenson,

Thank you for submitting your manuscript to PLOS ONE. After careful consideration, we feel that it has merit but does not fully meet PLOS ONE’s publication criteria as it currently stands. Therefore, we invite you to submit a revised version of the manuscript that addresses the points raised during the review process.

 As listed in the reviews, there were multiple concerns regarding this study as related to doses used, interpretation of results, and novelty of the results.  If you decide to resubmit this manuscript, please be sure to directly address these concerns.

We look forward to receiving your revised manuscript.

Kind regards,

R. Mark Wooten, Ph.D.

Academic Editor

PLOS ONE

Journal Requirements:

"Supported by NIH grant R03 AI133056 to B. Stevenson and W. Zückert, and NIH award numbers U54GM128729 and 2P20GM104360-06A1 to the University of North Dakota Genomics core. We thank Tatiana Castro-Padovani, Andrew Krusenstjerna, and Nerina Jusufovic for helpful comments on these studies and the manuscript."

"BS, WRZ: R03 AI133056, National Institutes of Health. The funders had no role in study design, data collection and analysis, decision to publish, or preparation of the manuscript.

CAB: U54GM128729 and 2P20GM104360-06A1, National Institutes of Health. The funders had no role in study design, data collection and analysis, decision to publish, or preparation of the manuscript."

Reviewers' comments:

Reviewer's Responses to Questions

**Comments to the Author**

1. Is the manuscript technically sound, and do the data support the conclusions?

Reviewer #1: Yes

Reviewer #2: No

2. Has the statistical analysis been performed appropriately and rigorously? 

Reviewer #1: I Don't Know

Reviewer #2: No

3. Have the authors made all data underlying the findings in their manuscript fully available?

Reviewer #1: Yes

Reviewer #2: Yes

4. Is the manuscript presented in an intelligible fashion and written in standard English?

Reviewer #1: Yes

Reviewer #2: Yes

5. Review Comments to the Author

Reviewer #1: This report by Saylor, et al. describes the gene expression changes associated with culture of Borrelia burgdorferi in sublethal doses of doxycycline versus amoxicillin. The results are indeed novel and serve to help explain antibiotic tolerance to doxycycline. This reviewer has no specific issues with the science, yet there are numerous areas that need improvement with regard to presentation and interpretation of the data.

Introduction: paragraph 4, lines 71-79 need revision. First, the sentence lines 71-73 should include references 36, 37, 38, 40, 41 and 42 and should read “..have also reported persistence of intact spirochetes or their components by tick acquisition and molecular detection.” Line 74-76 only cites a critical interpretation of the results, suggesting that antibiotic treatment is sufficient and that the persisting organisms are not viable. Also, it is antibiotic regimens, not regiments. Finally, the lines 78-79 are erroneous in that these 2 papers were not considered:

Caskey JR, Hasenkampf NR, Martin DS, Chouljenko VN, Subramanian R, Cheslock MA, Embers ME. The Functional and Molecular Effects of Doxycycline Treatment on Borrelia burgdorferi Phenotype. Front Microbiol. 2019 Apr 18;10:690. doi: 10.3389/fmicb.2019.00690. PMID: 31057493; PMCID: PMC6482230.

Feng J., Shi W., Zhang S., Zhang Y. (2015). Persister mechanisms in Borrelia burgdorferi: implications for improved intervention. Emerg. Microbes Infect. 4:e51. 10.1038/emi.2015.51

While the Caskey paper is cited later in the manuscript, it was published well before this was submitted.

An important aspect of the reported study (versus the 2 others) is that the antibiotic treatment concentration is much lower that the reported MIC in vitro, it does reflect a concentration that can be expected to be achieved in the serum of human patients. This can be added to the discussion, where references on antibiotic tolerance by B. burgdorferi and other bacteria is warranted (and absent).

Materials/Methods:

-the section on preparation of cultures (lines 102-110) only lists amoxicillin twice instead of doxycycline.

For the RNA extraction, bioanalyzer results: please describe what “adequacy” means in terms of the data obtained. Also, the methods list 3 cultures (biological replicates) used for each time point. Were each of these analyzed individually with transx sequencing, or were there technical replicates as well? Please add more detail on how the replicates were processed and determined to be significant using the stated software. When revising this section, consider whether or not someone could replicate the study with the level of detail provided.

Results:

-line 150 states a “1:00” dilution

-line 216 should read “Microscopic examination..”

-Figures 2 and 4 should be presented side-by-side and labeled by drug treatment.

-Figure 3 has no panel labels

Discussion:

Two major aspects were lacking in the discussion: (1) a comparison to previous published results in terms of study design, common findings, different findings and interpretation. The Feng, et al. paper should be included here; and (2) an assessment of how these findings relate to patient treatment, antibiotic tolerance and treatment failure with doxycycline. The caveat that host adaptation (by the spirochete) is lacking should be included as a study limitation as well.

Reviewer #2: The author worked on the antibiotic response by Borrelia burgdorferi and observed the transcriptional changes after 3 h and 24 h post-treatment with sub-lethal concentrations of doxycycline and amoxicillin. The project design indicates the author's lack of in-depth knowledge about the antibiotic response. I did not find any novelty in this study; the paper is poorly written, and the explanation of their finding is quite confusing. Though this work has potential and is on a medically relevant topic, overall, the authors have not met the criteria to publish this work. The work does not distinguish itself from a previously published work in 2019. Thus, as written, it does not appear to be new work. The 2019 article (DOI: 10.3389/fmicb.2019.00690) has far more experimental data, contains a model organism (mice), and conclusions are well supported.

Most importantly, this manuscript does not contradict the 2019 findings. Furthermore, the authors' data analysis and assertions lack the expected scientific justification and rigor. Therefore, I have no choice but to recommend the rejection of this manuscript.

Strength:

Lyme disease is a relevant topic, and transcriptional analysis of antibiotic-treated and untreated populations is a reasonable approach, though I do have concerns about the concentrations of antibiotics used.

I appreciate the presence of Table 1 in the manuscript. Too often in the literature are the number of genes listed but not the actual names (Locus), or they are hard to find (bared) in the supplemental material.

Limitation/weakness:

Major concerns:

Line 30-32. The major conclusion is that sublethal concentration of doxycycline leads to increased levels of proteins involved in translation. How is this a new finding?

Line 136-137. Why were sublethal concentrations of antibiotics used for this work? The authors made it clear that they were studying this bacterium because it causes Lyme disease, and they want to understand how it survives antibiotics. Lyme disease is treated with typically Lethal dosages of antibiotics. Therefore, it only seems logical to isolate RNA from antibiotic persisters as others have previously done (see DOI: 10.1038/s41598-021-85509-7).

Lines 201-202: I am concerned if we can consider this work new and relevant. Another group already used doxycycline and published their data. Ref 69 used 50 ug/ml of doxycycline at a higher antibiotic concentration than the authors used here, 2 ug/ml. Furthermore, based on 10.2147/IDR.S19201, doxycycline MBC is 25 ug/ml. Thus, Ref 69 work is more relevant to antibiotic concentrations that kill this bacterium and more medically relevant. In addition, the data from Ref 69 is available. The authors could have compared their findings to this work. If they found something different, that would be interesting. For this work to be relevant, the authors would need to do mutational studies (knockdown, knockout, overexpression, or point mutation studies) of the genes they identified. Then test how well the bacteria survive the antibiotics.

I am also confused about the statement in lines 201-202, "While our studies were in progress, another research group published RNA-Seq results of B. burgdorferi that had been cultured for 5 days in 50 μg/ml doxycycline [69]." Ref 69 was published in 2019, which is about 3 years ago. They also did this work with mice and thus had the addition of a model organism to support their results. Based on this information and without comparing the author's work to Ref 69, it is my opinion that the paper's finding relies solely on the results for Amoxicillin. The authors state that "amoxicillin did not lead to significant changes in levels of any bacterial transcript."

Line 32-34: The authors state that the "amoxicillin did not lead to significant changes in levels of any bacterial transcript." I question this finding. If the amoxicillin concertation is at the MIC or above, one would expect a change in the transcriptome. Otherwise, how are the cells surviving? The authors should have described how they survived, for example, a protein level response. All other reported antibiotic-challenge studies have seen a change in gene expression. For this to be believable, the authors would need to test different levels of amoxicillin concentrations. Alternatively and the best course of action, would be to use lethal amoxicillin concentrations like others have done.

Line 247-248: "However, our transcriptomic analyses indicate that the amoxicillin-induced morphological changes were not genetically encode." This is a bold statement that I did not find substantial evidence from this work to be supported. Here is why:

1. RNA-seq analysis generally relies on using a 2-fold-cutoff as the authors did here. However, it is possible that a 1.5 gene change can have a significant effect on cell physiology? The authors must titrate antibiotic concentrations and isolate RNA from them to make this claim with reasonable support. Alternatively, they could test at lethal (MBC) concentrations.

2. A heterogenous population will have high noise levels and high variations (e.g. the standard deviations or SEM would be large). Bacterial antibiotic persister populations and tolerant populations are heterogeneous. Some cells survive, and others die. This has been tested for nearly 80 years; see Biggers 1944 (https://doi.org/10.1016/S0140-6736(00)74210-3). These variations mean that differences may be hidden. One method to get around these variations is to eliminate non-persister cells by using lethal (MBC) concentrations of the antibiotic, as others have done.

This study lacks a clear goal and significance. The explanation of their finding is vague, the author has a considerable lack of knowledge in literature, and more importantly, there is no novelty in this study.

Bacteria evade antibiotic treatment by entering into a metabolically repressed state, which is known as persistence (https://doi.org/10.1038/s41579-019-0196-3). The persistence mechanisms are widely studied and found in almost all bacterial species including the minimal cell (Mycoplasma mycoides JCVI-Syn3B) (doi: 10.1016/j.isci.2021.102391). There are also several studies was done about the persister formation of Borrelia burgdorferi strain (doi: 10.1186/s13071-019-3495-7 ). Here, the author studied antibiotic response on Borrelia burgdorferi, but they did not mention anything about the persistence, which seems to be quite unusual. The author should mention antibiotic persistence, and explain whether their study shows any light to reveal the mechanism of antibiotic tolerance or persistence, which is one of the crucial reasons behind antibiotic resistance.

Since the author did not observe any transcriptional changes in bactericidal concentration, then how are lines 56-60 relevant to this study?

Lines 78-79 contradict lines 201-202. In lines 78-79, the author said no study has been done yet at molecular level to observe genetically encoded responses during antibiotic treatment of this strain. In line 201-202, they mentioned another study had been done to understand transcriptional responses at bactericidal concentration. Please clarify.

As written, the introduction is not useful and verbose. The significance of the work beyond the connection to Lyme disease is not clear. The significance (the REASON) for the study should be the main focus of the introduction. I highly suggest the author remove the first paragraph describing Lyme disease symptoms because it is unnecessary. Instead, the authors should focus more on the bigger picture of their study, like why the study of antibiotic response is important.

The author checked the MIC value by adding antibiotic treatment and counting the cells/mL under a microscope using a Petroff Hausser hemocytometer. However, the method is not entirely accurate since the author will count both live and dead cells at the same time, and there will be cells a different planes of the hemocytometer, so there is a high chance of miscalculation. The author did not validate their counts using CFU/ml. Although the generally accepted way of doing MIC and MBC tests is using optical density assay, CFU/mL agar plate assays, strip assays, or disk assay using Kirby-Bauer method. Thus, I am confused why the author did the MIC test in this way. Moreover, they did not show the MBC value, which is important to understand the level of antibiotics used. Therefore, the authors need to explain the reason for this unusual experimental design.

In Fig. 1, the author showed that cell growth was inhibited with 0.2 and 0.4 ug/mL of an antibiotic (doxycycline and amoxicillin). However, the growth inhibition was visible at 2 days, not at 24 h. In 24 h, the doxycycline showed a little bit of difference. However, without statistics (e.g. standard deviation, SEM, etc.), the reader does not know if they overlap. Based on Fig. 1, there is no difference with or without amoxicillin. Again, no statistics to support this claim. Instead of seeing very little inhibition or no inhibition at all at 24 h of antibiotic treatment, the author should/could have used other time points to isolate RNA. Why did they use 3 h and 24 h of antibiotic-treated cells? Is there any good explanation for that? Why did the author not go more extended time point? The author should address these questions in the discussion.

The author also mentioned that they did not see any significant changes in gene expression levels at 24 h with amoxicillin treatment. It seems most plausible the author did not see any significant change because there is no significant growth inhibition at that time point (24 h) with antibiotic treatment. The author should address this concern in the manuscript. I also believe the author should isolate RNA from longer antibiotic-treated cells where the inhibition is quite visible and statically significant with the control. Also, the author should add p-values for each time point in Fig. 1.

If the authors are unwilling to do RNA-seq for a longer time point with amoxicillin, I suggest the author remove this part from the paper. Because their experimental design has a huge flaw, and their explanation does not make any sense. I am adding here another reference (https://doi.org/10.1038/s41598-021-85509-7). They studied the transcriptional response on Ampicillin (beta-lactam antibiotic) treated E. coli cells and observed several genes are upregulated or downregulated at 3 h and 6 h of antibiotic treated cells. Adding the amoxicillin results and their hypothesis weakens the overall manuscript, so I highly suggest removing this part.

10. Fig. 3: At what time point during antibiotic-treated cells were considered for this microscopic analysis? Also, why do the authors show only one or two cells in one image rather than showing multiple cells? Do all the cells follow the same pattern during long-term Amoxicillin treatment? There is also a morphological difference in Fig. 3. D, E, and F. Why so? Could you hypothesize or find a reference for why? Please clarify.

Minor concerns:

Fig. 3 does not have any numbering on top of the figure. You must add letters (A, B, C…) called out in the legend to the figure.

Lines 112-120: Was rRNA removed from the samples? Though it is not stated in the Materials and Methods, I am assuming so because this is a typical step to save on cost and expression analysis time. If so, it should be stated and how.

The author should mention how many reads of each sample they used to analyze the RNA-seq data.

Line 244: Fig. 3A should be removed since the author only referred to amoxicillin-treated cultures. So, it should be Fig. 3D-F.

4. Line 107: Authors should mention the speed, time and temperature used for centrifugation to isolate the cells. If the author used 4˚C to centrifuge the cells, does this temperature change affects the transcriptional response?

Author Contributions is missing? Who did what on this manuscript?

Grammar and such needs major improvement. Here are some examples:

Line 27: Remove the comma from "profiles, in order"

Line 150: No spaces between bacteria/ml

Table 1: Capitalize Locus and Log2

Line 164: A space is needed between Fig. and 2

Fig.: Fig. is an abbreviation for figure, and there should be a period after Fig.

Add reference for line 47 and lines 50-55

6. PLOS authors have the option to publish the peer review history of their article (what does this mean?). If published, this will include your full peer review and any attached files.

Reviewer #1: No

Reviewer #2: No

---

## [Author Response · Author response to Decision Letter 0]

18 Jul 2022

Dear Dr. Wooten,

Our thanks to you and the two reviewers for your evaluation of our research and manuscript. We have extensively revised the manuscript in response to the reviewers’ concerns. In particular, Reviewer 2 appears to have overlooked several aspects of the initial manuscript, and we hope that the revisions make our points more clear.

Additionally, we have removed the Introduction section’s mention of apparent failures of antibiotics to clear B. burgdorferi infections from animals and human patients. Both reviewers interpreted our study as having direct impact on those apparent treatment failures, which we did not intend to imply. In response to their comments, we do address this issue at the end of the revised Conclusions section.

We addressed the reviewers’ comments as follows (our responses indicated by ***).

Reviewer 1:

This report by Saylor, et al. describes the gene expression changes associated with culture of Borrelia burgdorferi in sublethal doses of doxycycline versus amoxicillin. The results are indeed novel and serve to help explain antibiotic tolerance to doxycycline. This reviewer has no specific issues with the science, yet there are numerous areas that need improvement with regard to presentation and interpretation of the data.

*** Thank you

Introduction: paragraph 4, lines 71-79 need revision. First, the sentence lines 71-73 should include references 36, 37, 38, 40, 41 and 42 and should read “..have also reported persistence of intact spirochetes or their components by tick acquisition and molecular detection.” Line 74-76 only cites a critical interpretation of the results, suggesting that antibiotic treatment is sufficient and that the persisting organisms are not viable.

*** We have rearranged the order of the manuscript, and incorporated these suggestions into what is now the final paragraph of the manuscript (lines 289 - 298)

Also, it is antibiotic regimens, not regiments.

*** The typographical error has been fixed.

Finally, the lines 78-79 are erroneous in that these 2 papers were not considered: Caskey JR, Hasenkampf NR, Martin DS, Chouljenko VN, Subramanian R, Cheslock MA, Embers ME. The Functional and Molecular Effects of Doxycycline Treatment on Borrelia burgdorferi Phenotype. Front Microbiol. 2019 Apr 18;10:690. doi: 10.3389/fmicb.2019.00690. PMID: 31057493; PMCID: PMC6482230. Feng J., Shi W., Zhang S., Zhang Y. (2015). Persister mechanisms in Borrelia burgdorferi: implications for improved intervention. Emerg. Microbes Infect. 4:e51. 10.1038/emi.2015.51

*** Thank you for your suggestions. Both Caskey et al. and Feng et al. are discussed, with contrasts and comparisons to each other and to our results.

While the Caskey paper is cited later in the manuscript, it was published well before this was submitted.

*** This was a holdover from earlier versions of the manuscript. These studies were begun several years ago, as a Master student’s work. We have removed the confusing phrase.

An important aspect of the reported study (versus the 2 others) is that the antibiotic treatment concentration is much lower that the reported MIC in vitro, it does reflect a concentration that can be expected to be achieved in the serum of human patients. This can be added to the discussion, where references on antibiotic tolerance by B. burgdorferi and other bacteria is warranted (and absent).

*** The purpose of our study was to determine whether antibiotic stresses to B. burgdorferi result in transcriptional changes. This can only be addressed with living bacteria. Antibiotic concentrations above the MIC will kill substantial numbers of bacteria, and dead bacteria cannot respond to the stress. Moreover, RNA of dead bacteria is subject to degradation, which can complicate interpretation of results.

Materials/Methods:

-the section on preparation of cultures (lines 102-110) only lists amoxicillin twice instead of doxycycline.

*** Typographical error has been fixed.

For the RNA extraction, bioanalyzer results: please describe what “adequacy” means in terms of the data obtained.

*** We have re-written the method, and clarified that RNA extraction and library production was performed by ACGT Inc., according to their standard methods.

Also, the methods list 3 cultures (biological replicates) used for each time point. Were each of these analyzed individually with transx sequencing, or were there technical replicates as well?

*** Each culture was analyzed individually, without technical replicates.

 Please add more detail on how the replicates were processed and determined to be significant using the stated software. When revising this section, consider whether or not someone could replicate the study with the level of detail provided.

*** The methods are described herein, and in our previous publications, such that they can be readily replicated.

Results:

line 150 states a “1:00” dilution

*** Typo fixed.

-line 216 should read “Microscopic examination..”

Either “microscopical” or “microscopic” are acceptable English. We prefer “microscopical”.

-Figures 2 and 4 should be presented side-by-side and labeled by drug treatment.

*** Due to rearrangement, the previous Figure 4 is now Figure 3, and former Figure 3 is now Figure 4. The data in Figures 2 and (now) 3 report data from two distinct studies, and should remain as separate figures. 

-Figure 3 has no panel labels

*** Oops! The wrong version of this figure was originally submitted. This has been remedied.

Discussion:

Two major aspects were lacking in the discussion: (1) a comparison to previous published results in terms of study design, common findings, different findings and interpretation. The Feng, et al. paper should be included here;

*** Thank you for calling the Feng et al. paper to our attention. The initial manuscript compared our results with those of Caskey et al. We have now expanded to include comparisons and contrasts between our results and those of both Feng and Caskey. Notably, both Feng and Caskey incubated B. burgdorferi in 50 �g/ml doxycycline for 5 days, which is well above the MIC, and did not address whether or not their studied bacteria were metabolically active. In our studies, both antibiotics were used at sublethal concentrations, and both growth curves and microscopical analyses showed that the bacteria were metabolically active. We also note that the results of Feng and Caskey were distinct from each other, even though both used the same methods.

(2) an assessment of how these findings relate to patient treatment, antibiotic tolerance and treatment failure with doxycycline.

*** We have addressed these points in lines 289-298.

The caveat that host adaptation (by the spirochete) is lacking should be included as a study limitation as well.

*** We now bring this up in lines 290-293.

Reviewer 2:

The author worked on the antibiotic response by Borrelia burgdorferi and observed the transcriptional changes after 3 h and 24 h post-treatment with sub-lethal concentrations of doxycycline and amoxicillin. The project design indicates the author's lack of in-depth knowledge about the antibiotic response.

*** This statement, and other below, suggest that reviewer was confused about the goals of our study. We hope that the revised manuscript is more clear. 

I did not find any novelty in this study; the paper is poorly written, and the explanation of their finding is quite confusing.

*** These studies have never been performed before, to the best of our knowledge. We hope that the revised manuscript is easier to understand.

Though this work has potential and is on a medically relevant topic, overall, the authors have not met the criteria to publish this work. The work does not distinguish itself from a previously published work in 2019. Thus, as written, it does not appear to be new work.

*** The initial manuscript compared and contrasted with that 2019 work (Caskey et al.). There are substantial differences in the methods. Chiefly among these, Caskey et al. incubated B. burgdorferi for 5 days in 50 �g/ml doxycycline, which is well above the MIC for that antibiotic. In contrast, we incubated B. burgdorferi in 0.2 �g/ml doxycycline, which is below the MIC, and thus allowed the bacteria to remain metabolically active and, potentially, adjust transcription levels.

The 2019 article (DOI: 10.3389/fmicb.2019.00690) has far more experimental data, contains a model organism (mice), and conclusions are well supported.

*** The only mouse work of Caskey et al. involved them injecting bacteria (which had been cultured in 50 �g/ml doxycycline for 5 days) into mice. The result showed that some bacteria were not killed by that level of doxycycline. We do not debate that point.

Most importantly, this manuscript does not contradict the 2019 findings.

*** In the initial manuscript, and again in the revised version, we discussed differences in our results and those of Caskey et al. Primarily, Caskey et al. reported increases of 20 transcripts, all but one of which came from the Lyme spirochete’s resident cp32 prophages. The vast majority of prophage genes were not affected. The increased transcripts encode portal proteins of 4 different cp32 bacteriophages, and 3 different Erp lipoproteins. In contrast, we observed differential expression of 151 genes, with 143 upregulated and 8 downregulated. 53/151 differentially expressed genes are involved in protein synthesis, and account for nearly half (47%) of all genes annotated as belonging to the translation, ribosomal structure, and biogenesis pathway.

Furthermore, the authors' data analysis and assertions lack the expected scientific justification and rigor. Therefore, I have no choice but to recommend the rejection of this manuscript.

*** The reviewer did not support these assertions, so we cannot address their complaints.

Strength:

Lyme disease is a relevant topic, and transcriptional analysis of antibiotic-treated and untreated populations is a reasonable approach, though I do have concerns about the concentrations of antibiotics used.

*** The concentrations of antibiotics were appropriate for the questions at hand. We also note that reviewer 1 stated that the studied antibiotic concentrations are medically relevant.

I appreciate the presence of Table 1 in the manuscript. Too often in the literature are the number of genes listed but not the actual names (Locus), or they are hard to find (bared) in the supplemental material.

*** Thank you. We also find it frustrating when reports do not lay out RNA-Seq data in ways that are difficult for readers to follow.

Limitation/weakness:

Major concerns:

Line 30-32. The major conclusion is that sublethal concentration of doxycycline leads to increased levels of proteins involved in translation. How is this a new finding?

*** We are not aware of any publication that addresses this. The reviewer did not support their assertion that our findings are not new. 

Line 136-137. Why were sublethal concentrations of antibiotics used for this work?

*** Dead bacteria do not actively transcribe mRNAs. We wanted to understand how B. burgdorferi respond to antibiotic stresses, which requires use of sublethal concentrations of antibiotics.

The authors made it clear that they were studying this bacterium because it causes Lyme disease, and they want to understand how it survives antibiotics.

*** We hope that the revised manuscript makes it more clear that we were addressing how B. burgdorferi responds to antibiotic stresses, not how “it survives antibiotics”. Those are separate questions.

Lyme disease is treated with typically Lethal dosages of antibiotics. Therefore, it only seems logical to isolate RNA from antibiotic persisters as others have previously done (see DOI: 10.1038/s41598-021-85509-7).

*** The referenced paper is about E. coli in amoxicillin. Our studies indicated that B. burgdorferi does not undergo transcriptional alternations in response to amoxicillin stress. Use of higher doses would have killed the B. burgdorferi and, as noted above, dead bacteria do not produce mRNAs.

Lines 201-202: I am concerned if we can consider this work new and relevant.

*** As noted above, there are substantial differences between our studies and anything else that has been published before.

Another group already used doxycycline and published their data. Ref 69 used 50 ug/ml of doxycycline at a higher antibiotic concentration than the authors used here, 2 ug/ml. Furthermore, based on 10.2147/IDR.S19201, doxycycline MBC is 25 ug/ml. Thus, Ref 69 work is more relevant to antibiotic concentrations that kill this bacterium and more medically relevant.

*** Use of 50 �g/ml doxycycline would mean that the vast majority of bacteria are dead. We wanted to know how live bacteria respond to antibiotic stress, and so we used antibiotic concentrations that were stressful but not lethal. Microscopical examination revealed that bacteria in our studies were metabolically active at the times of harvest.

In addition, the data from Ref 69 is available. The authors could have compared their findings to this work. If they found something different, that would be interesting.

*** We discussed those differences in the initial manuscript, and include that discussion in the revised manuscript on lines 213-234 and 293-296.

For this work to be relevant, the authors would need to do mutational studies (knockdown, knockout, overexpression, or point mutation studies) of the genes they identified. Then test how well the bacteria survive the antibiotics.

*** We were not asking whether B. burgdorferi survive in antibiotics, but whether they possess genetically encoded responses to amoxicillin and doxycycline. Mutations of proteins that are involved with translation processes would not provide any insights.

*** We gently remind the reviewer that neither Caskey et al. nor Feng et al. performed any analyses of mutant B. burgdorferi.

I am also confused about the statement in lines 201-202, "While our studies were in progress, another research group published RNA-Seq results of B. burgdorferi that had been cultured for 5 days in 50 μg/ml doxycycline [69]." Ref 69 was published in 2019, which is about 3 years ago.

*** This was a holdover from earlier versions of the manuscript. These studies were begun several years ago, as a Master student’s work. We have removed the confusing phrase.

They also did this work with mice and thus had the addition of a model organism to support their results.

*** The only use of mice in Caskey et al. was to help show that B. burgdorferi were not all dead after 5 days in 50 ug/ml doxycycline.

Based on this information and without comparing the author's work to Ref 69, it is my opinion that the paper's finding relies solely on the results for Amoxicillin.

*** For the reasons discussed above, we reject this conclusion.

Line 32-34: The authors state that the "amoxicillin did not lead to significant changes in levels of any bacterial transcript." I question this finding. If the amoxicillin concertation is at the MIC or above, one would expect a change in the transcriptome.

*** The concentration of amoxicillin that we used caused stress to the bacteria, as evidenced by the reduced replication rate and formation of spheroplasts. The reviewer appears to be assuming that a change in transcript levels should occur, even though we found that there were no significant changes.

Otherwise, how are the cells surviving?

*** The bacteria survived because the level of amoxicillin was below the threshold for killing. 

The authors should have described how they survived, for example, a protein level response.

*** Our dose was below the MIC.

All other reported antibiotic-challenge studies have seen a change in gene expression. For this to be believable, the authors would need to test different levels of amoxicillin concentrations.

*** Our growth curve analyses indicated that B. burgdorferi were severely reduced in replication rate, but did continue to replicate at 0.2 ug/ml amoxicillin. Microscopical analyses indicated that they were metabolically active. Use of lower antibiotic concentrations would be less stressful, and we consider it unlikely that lower stress would provoke a greater response. Use of higher antibiotic concentrations would have killed the bacteria, and dead bacteria do not yield useful RNA-Seq data.

 Alternatively and the best course of action, would be to use lethal amoxicillin concentrations like others have done.

*** Dead bacteria do not actively transcribe RNA, so RNA-Seq analyses reveal only the transcripts that were produced before antibiotic was administered, with a complication of RNA degradation in the dead cells.

Line 247-248: "However, our transcriptomic analyses indicate that the amoxicillin-induced morphological changes were not genetically encode." This is a bold statement that I did not find substantial evidence from this work to be supported.

*** The data show otherwise.

Here is why: 1. RNA-seq analysis generally relies on using a 2-fold-cutoff as the authors did here. However, it is possible that a 1.5 gene change can have a significant effect on cell physiology?

*** We agree with the reviewer that fold-change cutoffs are arbitrary and can lead to misleading conclusions. To this end, we displayed our data so that the reader can see the potential contribution of those genes with statistically significant expression differences that don't make the somewhat arbitrary 2-fold cutoff. MA plots in Figs 2-3 distinguish those as yellow dots (denoted "significant no change (sigNC)"), whereas red/blue dots are those statistically significant genes that did meet our cutoff to be considered differentially expressed. As such, figure 3 shows that amoxicillin did not induce any statistically significant changes, regardless of the fold-change cutoff at any timepoint. The text has been modified to reflect this.

The authors must titrate antibiotic concentrations and isolate RNA from them to make this claim with reasonable support. Alternatively, they could test at lethal (MBC) concentrations.

*** These were addressed above.

2. A heterogenous population will have high noise levels and high variations (e.g. the standard deviations or SEM would be large).

*** This is good reason to use a stringent cutoff for data analysis.

Bacterial antibiotic persister populations and tolerant populations are heterogeneous. Some cells survive, and others die. This has been tested for nearly 80 years; see Biggers 1944 (https://doi.org/10.1016/S0140-6736(00)74210-3).

*** The cited paper dealt with penicillin treatment of staphylococcal infections. Staphylococcus spp. are distinct from Borrelia spp.

These variations mean that differences may be hidden. One method to get around these variations is to eliminate non-persister cells by using lethal (MBC) concentrations of the antibiotic, as others have done.

*** We addressed this above.

This study lacks a clear goal and significance.

*** We disagree. We hope that the revised manuscript clears up the reviewer’s apparent confusion.

The explanation of their finding is vague, the author has a considerable lack of knowledge in literature, and more importantly, there is no novelty in this study.

*** We disagree, as discussed above.

Bacteria evade antibiotic treatment by entering into a metabolically repressed state, which is known as persistence (https://doi.org/10.1038/s41579-019-0196-3).

*** We have included this reference (Balaban et al) in our revised manuscript. Among other things, the authors of that publication explicitly state that bacteria that do not die in bacteriostatic antibiotics (such as doxycycline) are not to be called “persisters”. The reason being that bacteria can survive bacteriostatic antibiotics without undergoing any biological changes.

The persistence mechanisms are widely studied and found in almost all bacterial species including the minimal cell (Mycoplasma mycoides JCVI-Syn3B) (doi: 10.1016/j.isci.2021.102391). There are also several studies was done about the persister formation of Borrelia burgdorferi strain (doi: 10.1186/s13071-019-3495-7 ). Here, the author studied antibiotic response on Borrelia burgdorferi, but they did not mention anything about the persistence, which seems to be quite unusual. The author should mention antibiotic persistence, and explain whether their study shows any light to reveal the mechanism of antibiotic tolerance or persistence, which is one of the crucial reasons behind antibiotic resistance.

*** Our studies revealed that exposure of B. burgdorferi to amoxicillin stress did not result in any significant transcriptional changes. This suggests that B. burgdorferi does not possess a mechanism to adapt to amoxicillin, and argues against the hypothesis that B. burgdorferi can produce true “persister” cells against amoxicillin.

*** Survival in doxycycline does not indicate production of “persister” bacteria. Indeed, Balaban et al. argue that discussion of persistence is not appropriate for tetracyclines and other bacteriostatic antibiotics.

*** The revised manuscript includes discussion of these points on lines 208-213.

Since the author did not observe any transcriptional changes in bactericidal concentration, then how are lines 56-60 relevant to this study?

*** We recognize that those lines were tangential to the report, and have removed them from the revised manuscript.

Lines 78-79 contradict lines 201-202. In lines 78-79, the author said no study has been done yet at molecular level to observe genetically encoded responses during antibiotic treatment of this strain. In line 201-202, they mentioned another study had been done to understand transcriptional responses at bactericidal concentration. Please clarify.

*** We have removed the sentence.

As written, the introduction is not useful and verbose. The significance of the work beyond the connection to Lyme disease is not clear. The significance (the REASON) for the study should be the main focus of the introduction. I highly suggest the author remove the first paragraph describing Lyme disease symptoms because it is unnecessary. Instead, the authors should focus more on the bigger picture of their study, like why the study of antibiotic response is important.

*** This is an opinion on writing style. We disagree with the reviewer’s opinion.

The author checked the MIC value by adding antibiotic treatment and counting the cells/mL under a microscope using a Petroff Hausser hemocytometer. However, the method is not entirely accurate since the author will count both live and dead cells at the same time, and there will be cells a different planes of the hemocytometer, so there is a high chance of miscalculation.

*** Dr. Zückert and I each have over 30 years of experience working with B. burgdorferi. The other authors also have numerous years of experience in this field. We know how to count B. burgdorferi with a hemocytometer.

The author did not validate their counts using CFU/ml.

*** Efficiency of plating for B. burgdorferi is generally less than 100%. Counting by microscopy with a hemocytometer is more accurate and is widely used in this field.

Although the generally accepted way of doing MIC and MBC tests is using optical density assay, CFU/mL agar plate assays, strip assays, or disk assay using Kirby-Bauer method. Thus, I am confused why the author did the MIC test in this way.

*** B. burgdorferi do not form lawns on the surfaces of agar plates, which precludes use of diffusion assays.

Moreover, they did not show the MBC value, which is important to understand the level of antibiotics used. Therefore, the authors need to explain the reason for this unusual experimental design.

*** MBC data have previously been determined. Our results were consistent with those data. Contrary to the reviewer’s opinion, there is nothing unusual about examining responses of bacteria to sublethal antibiotic stresses.

In Fig. 1, the author showed that cell growth was inhibited with 0.2 and 0.4 ug/mL of an antibiotic (doxycycline and amoxicillin). However, the growth inhibition was visible at 2 days, not at 24 h. In 24 h, the doxycycline showed a little bit of difference. However, without statistics (e.g. standard deviation, SEM, etc.), the reader does not know if they overlap. Based on Fig. 1, there is no difference with or without amoxicillin. Again, no statistics to support this claim.

*** We sought to examine stressed, but not dead, bacteria. The suggested statistical analyses are irrelevant.

Instead of seeing very little inhibition or no inhibition at all at 24 h of antibiotic treatment, the author should/could have used other time points to isolate RNA. Why did they use 3 h and 24 h of antibiotic-treated cells? Is there any good explanation for that? Why did the author not go more extended time point? The author should address these questions in the discussion.

*** This is addressed in lines 151-153 of the revised manuscript.

The author also mentioned that they did not see any significant changes in gene expression levels at 24 h with amoxicillin treatment. It seems most plausible the author did not see any significant change because there is no significant growth inhibition at that time point (24 h) with antibiotic treatment. The author should address this concern in the manuscript.

*** We noted blebbing and other signs of stress as a result of 24 h treatment with amoxicillin.

I also believe the author should isolate RNA from longer antibiotic-treated cells where the inhibition is quite visible and statically significant with the control.

*** Discussed above.

Also, the author should add p-values for each time point in Fig. 1.

*** Discussed above.

If the authors are unwilling to do RNA-seq for a longer time point with amoxicillin, I suggest the author remove this part from the paper.

*** Discussed above. 

Because their experimental design has a huge flaw, and their explanation does not make any sense. I am adding here another reference (https://doi.org/10.1038/s41598-021-85509-7). They studied the transcriptional response on Ampicillin (beta-lactam antibiotic) treated E. coli cells and observed several genes are upregulated or downregulated at 3 h and 6 h of antibiotic treated cells.

*** E. coli are not the same bacteria as B. burgdorferi. One cannot infer responses of B. burgdorferi by examining another species. 

 Adding the amoxicillin results and their hypothesis weakens the overall manuscript, so I highly suggest removing this part.

*** Discussed above.

10. Fig. 3: At what time point during antibiotic-treated cells were considered for this microscopic analysis?

*** The legend states 24 hours.

Also, why do the authors show only one or two cells in one image rather than showing multiple cells?

*** Bacteria were spread out, so 40x images generally included only a single bacterium. Lower magnifications make it difficult to clearly see bacteria.

*** We have revisited the study, and now morphological analyses of ca. 100 bacteria after 24 h in amoxicillin in lines 252-266.

*** To avoid distraction, we have omitted the effects of doxycycline on B. burgdorferi cell lengths. The figure (now Fig. 4) has been revised to show only a control bacterium and 3 representative amoxicillin-treated bacteria.

Do all the cells follow the same pattern during long-term Amoxicillin treatment? There is also a morphological difference in Fig. 3. D, E, and F. Why so? Could you hypothesize or find a reference for why? Please clarify. 

*** These are probably spheroplasts. We cite several previously published papers of B. burgdorferi that were incubated in sublethal concentrations of �-lactam antibiotics which showed similar morphologies.

Minor concerns:

Fig. 3 does not have any numbering on top of the figure. You must add letters (A, B, C…) called out in the legend to the figure.

*** Fixed.

Lines 112-120: Was rRNA removed from the samples? Though it is not stated in the Materials and Methods, I am assuming so because this is a typical step to save on cost and expression analysis time. If so, it should be stated and how.

*** Line 110 of the materials and methods section states that Zymo-Seq Ribofree Total RNA Library Kits were used.

The author should mention how many reads of each sample they used to analyze the RNA-seq data.

*** Those data are in the Supplemental Tables.

Line 244: Fig. 3A should be removed since the author only referred to amoxicillin-treated cultures. So, it should be Fig. 3D-F.

*** Panel 3A shows contrast between untreated and treated bacteria.

4. Line 107: Authors should mention the speed, time and temperature used for centrifugation to isolate the cells. If the author used 4˚C to centrifuge the cells, does this temperature change affects the transcriptional response?

*** Centrifugation conditions have been added to line 100. Chilling bacteria inhibits activities of enzymes that could degrade RNA. Our earlier studies of B. burgdorferi found that even 8 hours of changed temperature did not result in alterations (Stevenson et al., 1995, Infect. Immun. 63: 4535-4539, PMID: 7591099).

Author Contributions is missing? Who did what on this manuscript?

*** The journal does not appear to have a place for that.

Grammar and such needs major improvement. Here are some examples: Line 27: Remove the comma from "profiles, in order" Line 150: No spaces between bacteria/ml Table 1: Capitalize Locus and Log2 Line 164: A space is needed between Fig. and 2 Fig.: Fig. is an abbreviation for figure, and there should be a period after Fig. Add reference for line 47 and lines 50-55

*** Thank you.

---

## [Editor Report · Decision Letter 1]

23 Aug 2022

Borrelia burgdorferi, the Lyme disease spirochete, possesses genetically-encoded responses to doxycycline, but not to amoxicillin

PONE-D-22-10059R1

Dear Dr. Stevenson,

We’re pleased to inform you that your manuscript has been judged scientifically suitable for publication and will be formally accepted for publication once it meets all outstanding technical requirements.

Kind regards,

Nikhat Parveen, Ph.D.

Academic Editor

PLOS ONE

Additional Editor Comments (optional):

Dear Dr. Stevenson,

Thank you for submitting your manuscript to PloS One (Number PONE-D-22-10059R1). After careful evaluation of your response in revised manuscript, we think you have addressed reviewers’ concerns and comments adequately. Therefore, we are pleased to inform you that your manuscript " Borrelia burgdorferi, the Lyme disease spirochete, possesses genetically-encoded responses to doxycycline, but not to amoxicillin" has been judged scientifically suitable and thus, will be formally accepted for publication in PloS One very soon.

In the near future, you will receive an e-mail containing information on any amendments required prior to publication. When all required modifications have been addressed, you will receive a formal acceptance letter and your manuscript will proceed to our production department and be scheduled for publication.

Kind regards,

Nikhat Parveen, Ph.D.

Academic Editor

PLOS ONE
---

## [Editor Report · Acceptance letter]

22 Sep 2022

PONE-D-22-10059R1 

*Borrelia burgdorferi*, the Lyme disease spirochete, possesses genetically-encoded responses to doxycycline, but not to amoxicillin 

Dear Dr. Stevenson:

I'm pleased to inform you that your manuscript has been deemed suitable for publication in PLOS ONE. Congratulations! Your manuscript is now with our production department. 

Kind regards, 

on behalf of

Dr. Nikhat Parveen 

Academic Editor

PLOS ONE